# A revised biosynthetic pathway for the cofactor $F_{420}$ in prokaryotes

Ghader Bashiri [1], James Antoney[2,3], Ehab N.M. Jirgis[1], Mihir V. Shah[2], Blair Ney[2,3], Janine Copp[4], Stephanie M. Stuteley[1], Sreevalsan Sreebhavan[5], Brian Palmer[5], Martin Middleditch[1], Nobuhiko Tokuriki [4], Chris Greening [2,6], Colin Scott [2], Edward N. Baker[1] & Colin J. Jackson [2,3]

Cofactor $F_{420}$ plays critical roles in primary and secondary metabolism in a range of bacteria and archaea as a low-potential hydride transfer agent. It mediates a variety of important redox transformations involved in bacterial persistence, antibiotic biosynthesis, pro-drug activation and methanogenesis. However, the biosynthetic pathway for $F_{420}$ has not been fully elucidated: neither the enzyme that generates the putative intermediate 2-phospho-L-lactate, nor the function of the FMN-binding C-terminal domain of the γ-glutamyl ligase (FbiB) in bacteria are known. Here we present the structure of the guanylyltransferase FbiD and show that, along with its archaeal homolog CofC, it accepts phosphoenolpyruvate, rather than 2-phospho-L-lactate, as the substrate, leading to the formation of the previously uncharacterized intermediate dehydro-$F_{420}$-0. The C-terminal domain of FbiB then utilizes $FMNH_2$ to reduce dehydro-$F_{420}$-0, which produces mature $F_{420}$ species when combined with the γ-glutamyl ligase activity of the N-terminal domain. These new insights have allowed the heterologous production of $F_{420}$ from a recombinant $F_{420}$ biosynthetic pathway in *Escherichia coli*.

[1] School of Biological Sciences and Maurice Wilkins Centre for Molecular Biodiscovery, The University of Auckland, Auckland 1010, New Zealand. [2] Synthetic Biology Future Science Platform, CSIRO Land and Water, Canberra 2601 ACT, Australia. [3] Research School of Chemistry, Australian National University, Acton 2601 ACT, Australia. [4] Michael Smith Laboratories, University of British Columbia, Vancouver, BC V6T 1Z4, Canada. [5] Auckland Cancer Society Research Centre, Faculty of Medical and Health Sciences, The University of Auckland, Auckland 1010, New Zealand. [6] School of Biological Sciences, Monash University, Clayton 3800 VIC, Australia. These authors contributed equally: Ghader Bashiri, James Antoney. Correspondence and requests for materials should be addressed to G.B. (email: g.bashiri@auckland.ac.nz) or to C.S. (email: colin.scott@csiro.au) or to C.J.J. (email: colin.jackson@anu.edu.au)

Cofactor $F_{420}$ is a deazaflavin that acts as a hydride carrier in diverse redox reactions in both bacteria and archaea[1,2]. While $F_{420}$ structurally resembles the flavins flavin mononucleotide (FMN) and flavin adenine dinucleotide (FAD), it functions as an obligate two-electron hydride carrier and hence is functionally analogous to the nicotinamides $NAD^+$ and $NADP^+$ [3]. The lower reduction potential of the $F_{420}$, relative to the flavins, results from the substitution of N5 of the isoalloxazine ring in the flavins for a carbon in $F_{420}$[4,5]. Originally characterized from methanogenic archaea in 1972[4,5], $F_{420}$ is an important catabolic cofactor in methanogens and mediates key one-carbon transformations of methanogenesis[6]. $F_{420}$ has since been shown to be synthesized in a range of archaea and bacteria[1,2,7,8]. In *Mycobacterium tuberculosis*, the causative agent of tuberculosis, $F_{420}$ has been shown to contribute to persistence[9,10] and to activate the new clinical antitubercular prodrugs delamanid and pretomanid[11]. There are also growing numbers of natural products that have been shown to be synthesized through $F_{420}$-dependent pathways, including tetracyclines[12], lincosamides[13], and thiopeptides[14]. $F_{420}$-dependent enzymes have also been explored for bioremediation and biocatalytic applications[15,16].

The currently accepted $F_{420}$ biosynthetic pathway consists of two branches[2] (Fig. 1). In the first branch, tyrosine is condensed with 5-amino-6-ribitylamino-2,4[1H,3H]-pyrimidinedione from the flavin biosynthetic pathway to generate the deazaflavin chromophore Fo (7,8-didemethyl-8-hydroxy-5-deazariboflavin) via the activity of the two-domain Fo synthase FbiC, or the CofG/H pair (where "Fbi" refers to mycobacterial proteins and "Cof" refers to archaeal homologs). In the second branch, it has been hypothesized that a 2-phospho-L-lactate guanylyltransferase (CofC in archaea and the putative enzyme FbiD in bacteria) catalyzes the guanylylation of 2-phospho-L-lactate (2-PL) using guanosine-5′-triphosphate (GTP), yielding L-lactyl-2-diphospho-5′-guanosine (LPPG)[17]. The two branches then merge at the reaction catalyzed by the transferase FbiA/CofD, where the 2-phospho-L-lactyl moiety of LPPG is transferred to Fo, forming $F_{420}$-0[18,19]. Finally, the γ-glutamyl ligase (FbiB/CofE) catalyzes the poly-glutamylation of $F_{420}$ to generate mature $F_{420}$, with poly-γ-glutamate tail lengths of ~2–8, depending on species[20,21].

There are three aspects of the $F_{420}$ biosynthetic pathway that require clarification. First, the metabolic origin of 2-PL, the proposed substrate for CofC, is unclear. It has been assumed that a hypothetical kinase (designated CofB) phosphorylates L-lactate to produce 2-PL[22]. However, no such enzyme has been identified in bacteria or archaea, and our genomic analysis of $F_{420}$ biosynthesis operons has failed to identify any candidate enzymes with putative L-lactate kinase activity[2]. Second, the existence of FbiD has only been inferred through bioinformatics and genetic knockout studies and the enzyme has not been formally characterized[23,24]. Finally, the bacterial γ-glutamyl ligase FbiB is a two-domain protein[20], in which the N-terminal domain is homologous to other $F_{420}$-γ-glutamyl ligases (including the archaeal equivalent, CofE) and the C-terminal domain adopts an FMN-binding nitroreductase (NTR) fold[20]. Although both domains are required for full γ-glutamyl ligase activity, no function has been associated with either the C-terminal domain or the FMN cofactor, given no redox reactions are known to be involved in $F_{420}$ biosynthesis.

Here we demonstrate that 2-PL is not required for $F_{420}$ biosynthesis in prokaryotes and instead phosphoenolpyruvate (PEP), an abundant intermediate of glycolysis and gluconeogenesis, is incorporated into $F_{420}$. Mass spectrometry (MS) and protein crystallography are used to demonstrate that PEP guanylylation is catalyzed by the FbiD/CofC enzymes that were previously thought to act upon 2-PL. In bacteria, the incorporation of PEP in the pathway results in the production of the previously undetected intermediate dehydro-$F_{420}$-0, which we identified by MS. We then showed, with the help of ligand docking, that this intermediate is then reduced by the C-terminal domian and poly-glutamylated by the N-terminal domain. These findings result in a substantially revised pathway for $F_{420}$ biosynthesis and have allowed us to heterologously express a functional $F_{420}$ biosynthetic pathway in *Escherichia coli*, an organism that does not normally produce $F_{420}$, at levels comparable to some native $F_{420}$-producing organisms.

## Results

**FbiD/CofC accept PEP, rather than 2-PL.** The archaeal enzyme CofC has previously been suggested to catalyze the guanylylation of 2-PL to produce LPPG during $F_{420}$ biosynthesis (Fig. 2a)[25]. Another study, using transposon mutagenesis, has shown that MSMEG_2392 of *Mycobacterium smegmatis* is essential in the biosynthesis of $F_{420}$ from Fo[23]. We have recently shown that homologs of this gene have sequence homology to CofC and belong to operons with other validated $F_{420}$ biosynthetic genes in a wide variety of bacteria[2]. In keeping with the bacterial naming system, we refer to this enzyme as FbiD. To test the function of this putative bacterial FbiD, we cloned the homologous Rv2983 gene from *M. tuberculosis*[26] into a mycobacterial expression vector and purified heterologously expressed *Mycobacterium tuberculosis* FbiD (*Mtb*-FbiD) from *M. smegmatis* mc²4517 host cells. We also expressed and purified *Mtb*-FbiA (the enzyme thought to transfer the 2-phospho-L-lactyl moiety of LPPG to Fo to produce $F_{420}$-0)[18] to use in coupled high-performance liquid chromatography-MS (HPLC-MS) enzymatic assays with *Mtb*-FbiD. Surprisingly, we found that when *Mtb*-FbiD and *Mtb*-FbiA were included in an assay with 2-PL, GTP (or ATP), and Fo, no product was formed (Fig. 2b). We then tested whether *Mtb*-FbiA and CofC from *Methanocaldococcus jannaschii* (*Mj*-CofC) could catalyze $F_{420}$-0 formation under the same conditions, which again yielded no product (Fig. 2b).

Although 2-PL is hypothesized to be an intermediate in $F_{420}$ biosynthesis, this has never been experimentally confirmed in bacteria. Additionally, no enzyme capable of phosphorylating L-lactate to 2-PL has been identified in $F_{420}$-producing organisms, despite considerable investigation[22]. 2-PL has been little studied as a metabolite and is only known to occur as a by-product of pyruvate kinase activity[27]. 2-PL has not been implicated as a substrate in any metabolic pathway outside the proposed role in $F_{420}$ biosynthesis; rather, it has been shown in vitro to inhibit several enzymes involved in glycolysis and amino acid biosynthesis[28–30]. Our inability to detect activity with 2-PL led us to consider alternative metabolites that could potentially substitute for 2-PL, including the structurally analogous and comparatively abundant molecule PEP[31] (Fig. 2a).

While there was no activity when 2-PL was used in the FbiD/CofC:FbiA-coupled assays, when these enzymes were incubated with PEP, GTP (or ATP), and Fo, a previously unreported intermediate in the $F_{420}$ biosynthesis pathway, which we term "dehydro-$F_{420}$-0," was produced (Fig. 2b). The identity of this compound, which is identical to $F_{420}$-0 except for a methylene group in place of the terminal methyl group, was verified by MS/MS (Fig. 2c). The only difference that we observed between the activities of *Mtb*-FbiD and *Mj*-CofC was that while *Mtb*-FbiD exclusively utilizes GTP to produce dehydro-$F_{420}$-0, *Mj*-CofC can also catalyze the reaction with ATP, albeit to a lesser extent (Fig. 2b). Interestingly, in our experiments the FbiD/CofC enzymes were only active in the presence of FbiA. This was not unexpected given that the inferred intermediate (enolpyruvyl-diphospho-5′-guanosine (EPPG)) is expected to be unstable[22] (Fig. 2a).

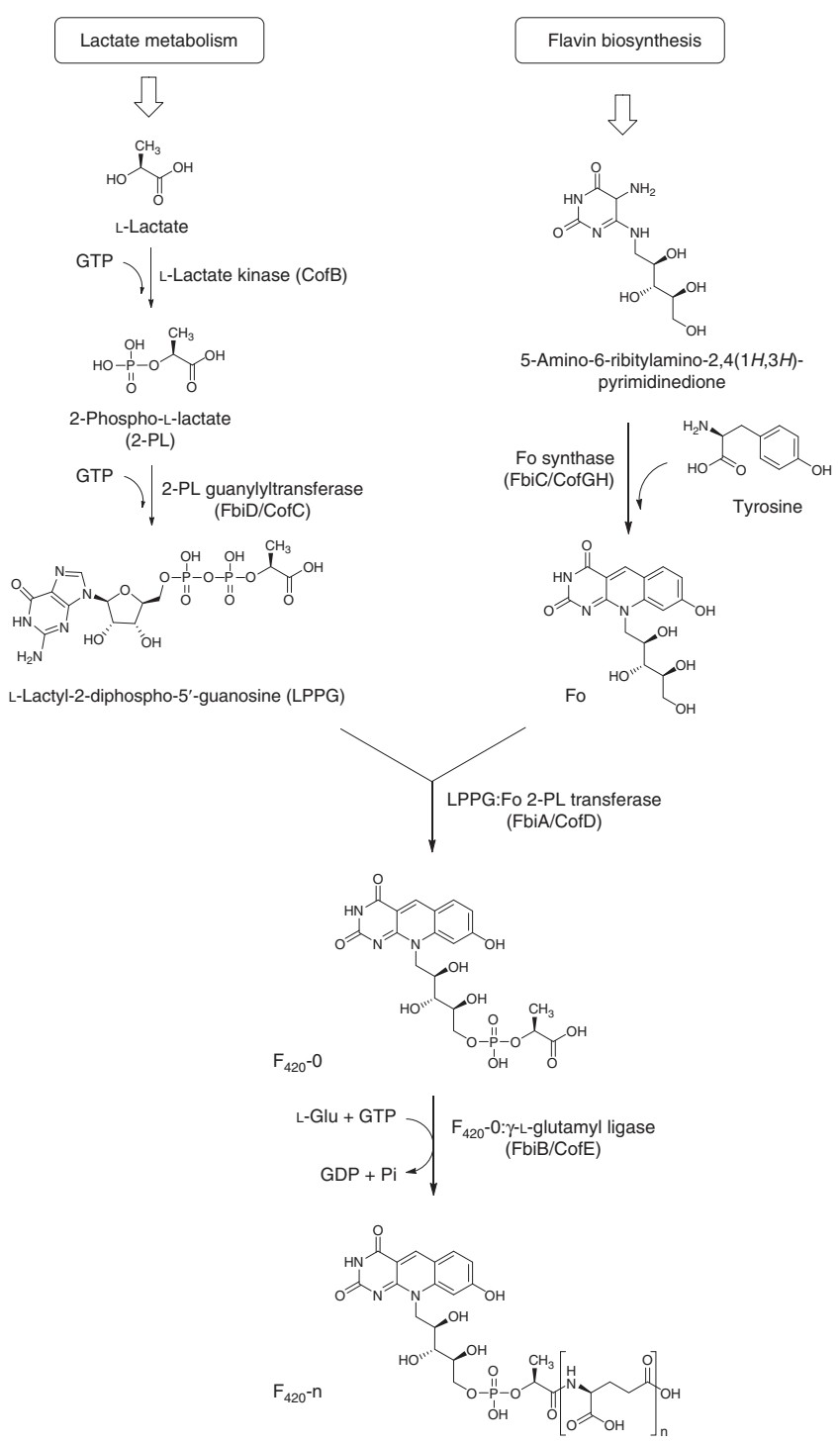

**Fig. 1** The currently accepted bacterial $F_{420}$ biosynthesis pathway. The pathway is composed of two branches. In the first branch tyrosine is condensed with the flavin biosynthesis intermediate 5-amino-6-ribitylamino-2,4-(1H,3H)-pyrimidinedione by FbiC/CofGH to produce the riboflavin level chromophore Fo. In the second branch, L-lactic acid is phosphorylated in a guanosine-5′-triphosphate (GTP)-dependent manner by the hypothetical L-lactate kinase CofB to produce 2-phospho-L-lactate (2-PL). This compound is subsequently guanylylated by FbiD/CofC to produce the unstable intermediate L-lactyl-2-diphospho-5′-guanosine (LPPG). The two branches merge as the 2-phospho-L-lactyl moiety of LPPG is transferred to Fo through the action of FbiA/CofD to produce $F_{420}$-0. Mature $F_{420}$ is then produced through glutamylation of $F_{420}$-0 by FbiB/CofE. "Fbi" refers to bacterial proteins, whereas "Cof" represents archaeal ones

To understand the molecular basis of PEP recognition by *Mtb*-FbiD, we crystallized the protein and solved the structure by selenium single-wavelength anomalous diffraction (Se-SAD), and then used this selenomethionine-substituted structure to obtain the native FbiD structure by molecular replacement. The latter was then refined at 1.99 Å resolution ($R/R_{free} = 0.19/0.22$) (Table 1). As expected, *Mtb*-FbiD adopts the same MobA-like nucleoside triphosphate transferase family protein fold as CofC: a

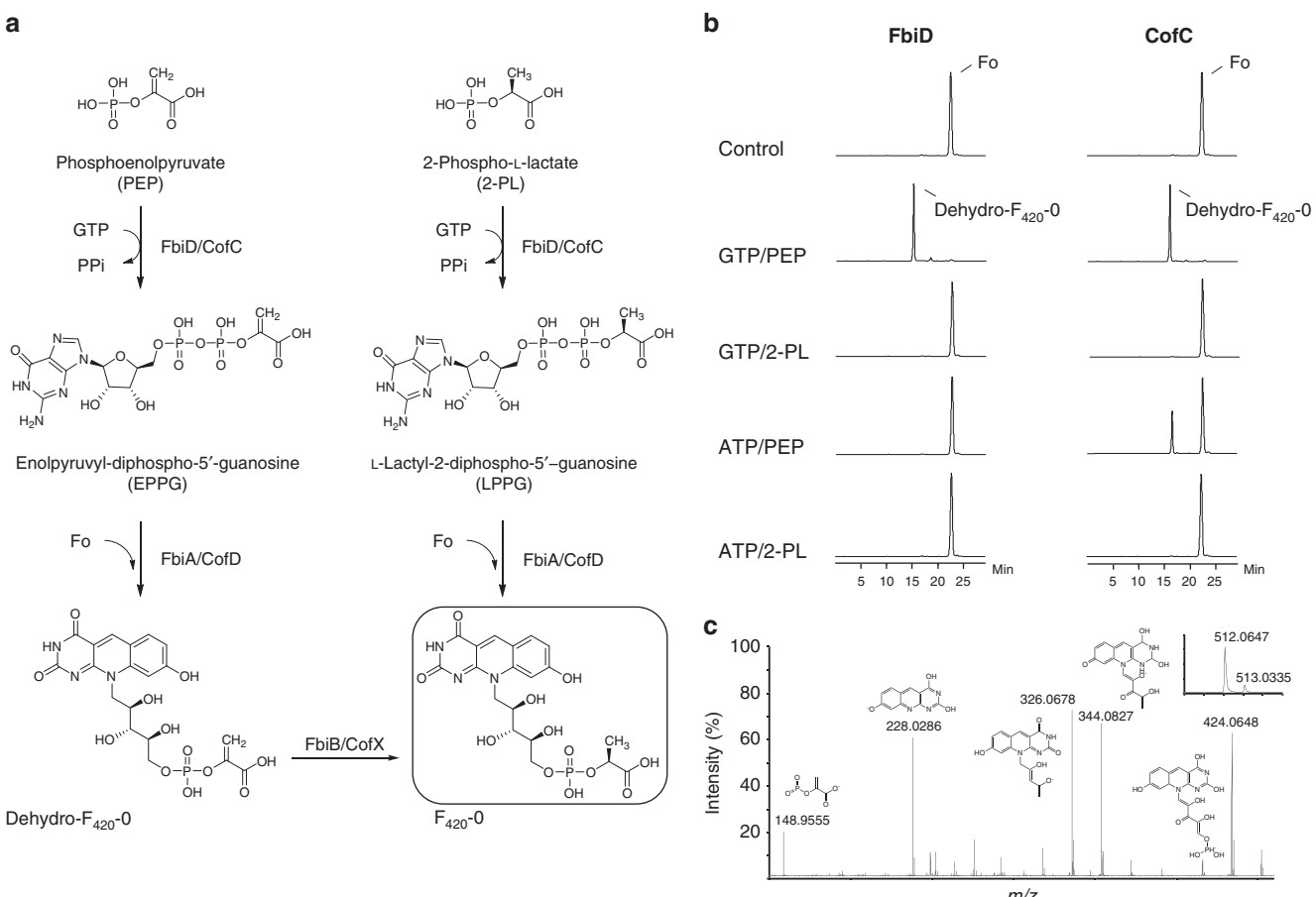

**Fig. 2** Phosphoenolpyruvate (PEP) is an intermediate in the formation of dehydro-$F_{420}$-0. **a** Production of $F_{420}$-0 in our revised biosynthesis pathway (left) compared to the currently accepted pathway (right). **b** Coupled-reaction high-performance liquid chromatography (HPLC) assays showing that both *Mtb*-FbiD and CofC from *Methanocaldococcus jannaschii* (*Mj*-CofC) enzymes use PEP to produce dehydro-$F_{420}$-0. **c** Tandem mass spectral identification of dehydro-$F_{420}$-0. Tandem mass spectrometry (MS/MS) fragmentation of dehydro-$F_{420}$-0, showing fragment ions with their corresponding structures. The inset displays the observed spectrum of the parent molecule (expected monoisotopic $m/z$ 512.0711 $[M - H]^-$)

central 7-stranded β-sheet (six parallel strands and one antiparallel), with α-helices packed on either side (Fig. 3a). However, *Mtb*-FbiD lacks the protruding hairpin that is important for dimer formation in CofC[32]. Superposition of CofC from *Methanosarcina mazei* (PDB code 2I5E) onto *Mtb*-FbiD gives a root mean square difference of 1.85 Å over 181 Cα atoms, with 25.4% sequence identity, establishing clear structural homology (Fig. 3c).

We also soaked PEP into pre-formed FbiD crystals to obtain an FbiD-PEP complex (2.18 Å resolution, $R/R_{free} = 0.22/0.26$). FbiD has a cone-shaped binding cleft with a groove running across the base of the cone, formed by the C-terminal end of the central β-sheet (Fig. 3a). PEP binds in the cleft with its phosphate group anchored through two $Mg^{2+}$ ions to three acidic side chains (D116, D188, and D190) (Fig. 3b). This three-aspartate motif is conserved amongst FbiD homologs as shown by a multiple sequence alignment (Supplementary Fig. 1). The PEP carboxylate group is hydrogen bonded to the hydroxyl group of S166 and the main chain NH groups of T148 and G163. All PEP binding residues are conserved in the CofC protein of *M. mazei* (PDB code 2I5E) (Supplementary Fig. 1), consistent with the enzymatic assays that showed PEP is the substrate for archaeal CofC, as well as FbiD. The reason that FbiD/CofC is active with PEP rather than 2-PL is most likely a consequence of the different stereochemistry of 2-PL (compared with the planar geometry of

PEP) preventing a necessary structural rearrangement, or the binding of GTP, which would be required to attain a productive transition state, as suggested for PEP carboxykinase[33]. In the GTP-bound structure of *E. coli* MobA[34] (PDB code 1FRW), GTP binds in a characteristic surface groove, providing a structural framework for substrate binding and catalysis. In our enzyme assays, neither FbiD nor CofC showed activity in the absence of FbiA. Moreover, we did not observe GTP binding in either our co-crystallization or differential scanning fluorimetry experiments. We speculate that the GTP binding site is not formed until FbiD/CofC interacts with FbiA/CofD, enabling catalysis to proceed through to formation of dehydro-$F_{420}$-0. This may provide an advantage by producing EPPG/EPPA only when both proteins are available, thereby overcoming the issue of intermediate instability.

**The C-terminal domain of FbiB reduces dehydro-$F_{420}$-0 to $F_{420}$-0.** Dehydro-$F_{420}$-0 would yield $F_{420}$-0 upon reduction of the terminal methylene double bond. However, no masses corresponding to $F_{420}$-0 were identified in any of the liquid chromatography-MS (LC-MS) traces from the FbiD:FbiA-coupled assays, suggesting that an enzyme other than FbiD or FbiA catalyzes dehydro-$F_{420}$-0 reduction. We have previously shown that full-length FbiB consists of two domains: an N-terminal

domain that is homologous to the archaeal γ-glutamyl ligase CofE[21,35], and a C-terminal domain of the NTR fold[36] that binds to FMN and has no known function, but is essential for extending the poly-γ-glutamate tail[20].

We tested whether FbiB could use dehydro-$F_{420}$-0 as a substrate with a three enzyme assay in which FbiB and L-glutamate were added to the FbiD:FbiA-coupled assay. *Mtb*-FbiB was observed to catalyze the addition of L-glutamate residues to dehydro-$F_{420}$-0, forming dehydro-$F_{420}$ species with one ($[M + H]^+$, *m/z* of 643.40) and two ($[M + H]^+$, *m/z* of 772.40) glutamate residues (Supplementary Fig. 2). We then tested the hypothesis that the orphan function of the FMN-binding C-terminal domain could in fact be a dehydro-$F_{420}$-0 reductase. We used a four-enzyme in vitro assay where *E. coli* NAD(P)H:flavin oxidoreductase[37] (Fre), FMN, NADH, and 10

mM dithiothreitol (to maintain reducing conditions and generate reduced $FMNH_2$) were added to the FbiD:FbiA:FbiB assay and the reaction was performed in anaerobic conditions (to prevent reaction of $FMNH_2$ with oxygen). We found that $F_{420}$-1, that is, the fully reduced and mature glutamylated cofactor, was produced, but only in the presence of both FbiB and Fre/$FMNH_2$ (Fig. 4a). Thus, dehydro-$F_{420}$-0 is a bona fide metabolic intermediate that can be converted to mature $F_{420}$ by FbiB in an $FMNH_2$-dependent fashion. These results demonstrate that bacterial FbiB is a bifunctional enzyme, functioning as a dehydro-$F_{420}$-0 reductase and as a γ-glutamyl ligase (Fig. 4b).

When the previously published crystal structures of *Mtb*-FbiB are analyzed in the context of these results, the molecular basis for this activity becomes clear. Crystal structures of the *Mtb*-FbiB C-terminal domain with $F_{420}$ bound (PDB ID: 4XOQ) and FMN bound (PBD ID: 4XOO) have been solved[20], and when these are overlaid it is apparent that the FMN molecule is ideally situated to transfer a hydride to the terminal methylene of dehydro-$F_{420}$-0 (assuming dehydro-$F_{420}$-0 binds in a similar fashion to $F_{420}$). Interestingly, the phospholactyl group of $F_{420}$ appears to be disordered in these crystal structures, suggesting it may adopt multiple conformations. To test this, we docked dehydro-$F_{420}$-0 into the $FMNH_2$-bound structure and performed an energy minimization using the OPLS3e forcefield to allow them to attain a low-energy conformation. This forcefield was chosen based on its improved torsional parameters for small molecules and more accurate descriptions of the potential energy surfaces of ligands, allowing for improved docking poses over other forcefields[38]. The results show that in this ternary complex the two molecules can adopt ideal positions and orientations for the reduction of dehydro-$F_{420}$-0 (Fig. 4c). The methylene group of dehydro-$F_{420}$-0 is accommodated by a small hydrophobic pocket mostly comprised of P289 and M372 allowing it to be positioned above the N5 atom of $FMNH_2$, in a plausible Michaelis complex for hydride transfer (Fig. 4c). We therefore suggest that the phosphoenolpyruvyl (analogous to the phospholactyl) group of dehydro-$F_{420}$-0 most likely samples conformations within this pocket where it can be reduced.

Interestingly, the γ-glutamyl ligase CofE from archaea is a single domain enzyme; there is no homology to the C-terminal NTR-fold domain of FbiB. In an analogous situation, Fo synthesis is performed by two single domain enzymes in archaea, CofH, and CofG[2], whereas in bacteria this reaction is catalyzed by a two-domain protein, FbiC (with N- and C-terminal domains homologous to CofH and CofG, respectively). Previous analysis

**Table 1 Data collection, processing, and refinement statistics**

|  | SeMet-FbiD[a] | Apo-FbiD[a] (PDB 6BWG) | FbiD-PEP[a] (PDB 6BWH) |
|---|---|---|---|
| **Data collection** | | | |
| Space group | C222₁ | C222₁ | C222₁ |
| Cell dimensions | | | |
| *a*, *b*, *c* (Å) | 66.56, 110.79, 167.69 | 66.54, 109.05, 166.49 | 66.17, 110.44, 165.91 |
| *α*, *β*, *γ* (°) | 90, 90, 90 | 90, 90, 90 | 90, 90, 90 |
| Resolution (Å) | 47.17–2.33 (2.41–2.33) | 46.92–1.99 (2.04–1.99) | 46.84–2.18 (2.25–2.18) |
| $R_{merge}$ | 0.13 (0.37) | 0.354 (2.71) | 0.213 (2.93) |
| $I / \sigma I$ | 21.3 (8.1) | 8.8 (1.2) | 14.5 (1.0) |
| $CC_{1/2}$ | 0.995 (0.97) | 0.994 (0.321) | 0.998 (0.376) |
| Completeness (%) | 99.91 (99.3) | 99.85 (96.9) | 100.0 (99.7) |
| Redundancy | 14.6 (14.2) | 14.8 (13.5) | 14.7 (13.2) |
| **Refinement** | | | |
| Resolution (Å) | | 83.25–1.99 | 82.95–2.18 |
| No. reflections | | 621278 (38169) | 471504 (35875) |
| $R_{work}/R_{free}$ | | 19.1/22.8 | 22.8/26.2 |
| No. atoms | | | |
| Protein | | 4969 | 4643 |
| Ligand/ion | | – | 36 |
| Water | | 544 | 180 |
| *B*-factors | | | |
| Protein | | 11.98 | 24.12 |
| PEP ($n = 3$) | | – | 48.78 |
| $Mg^{2+}$ ($n = 6$) | | – | 44.33 |
| Water | | 33.18 | 48.46 |
| R.m.s. deviations | | | |
| Bond lengths (Å) | | 0.011 | 0.01 |
| Bond angles (°) | | 1.416 | 1.383 |

Values within parentheses are for highest-resolution shell
[a]Number of crystals = 1

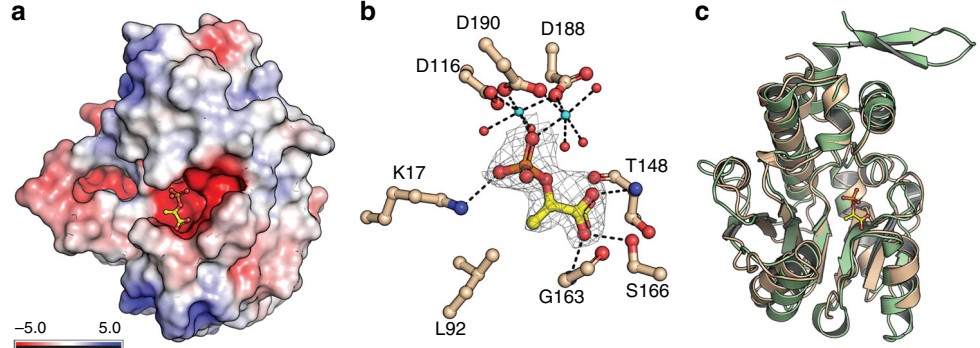

**Fig. 3** Crystal structure of *Mtb*-FbiD. **a** Electrostatic surface representation of the *Mtb*-FbiD structure in complex with phosphoenolpyruvate (PEP), shown as a ball-and-stick model. **b** The phosphate group of PEP binds to three aspartic acid side chains through two $Mg^{2+}$ ions (shown in cyan). PEP is shown in $2F_o - F_c$ omit density contoured at $2.0\sigma$, and drawn as ball-and-stick model. Water molecules are shown as red spheres and hydrogen bond interactions are outlined as dashed lines. **c** Superposition of *Mtb*-FbiD (wheat ribbon) and *Methanocaldococcus jannaschii* (*Mj*-CofC) (green ribbon), indicating 1.85 Å root mean square difference (rmsd) over 181 superimposed Cα. PEP is shown as a ball-and-stick model

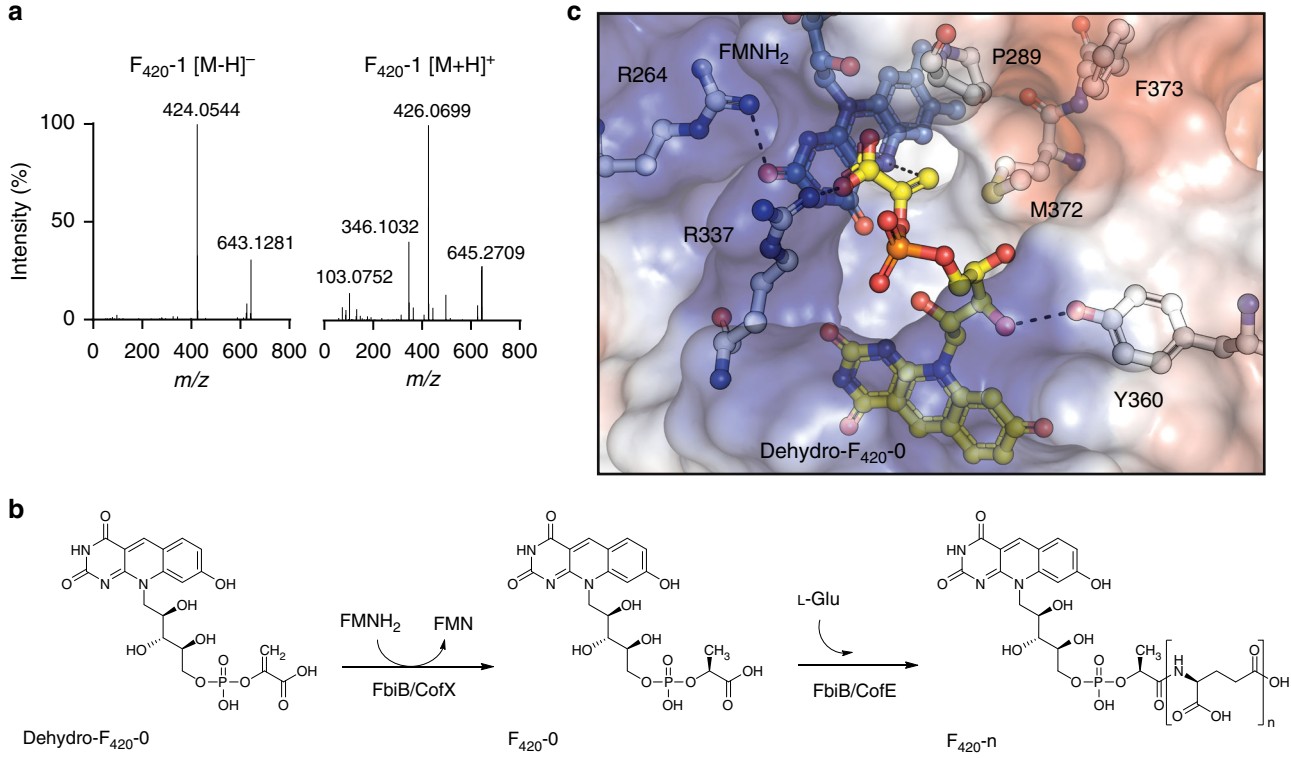

**Fig. 4** *Mtb*-FbiB catalyzes reduction of dehydro-$F_{420}$-0. **a** $F_{420}$-1 is produced in FbiD:FbiA:FbiB coupled assays in the presence of Fre/$FMNH_2$ and L-glutamate. Tandem mass spectrometry (MS/MS) confirmation of $F_{420}$-1 in both negative (643.12811, $[M − H]^−$) and positive (645.27094, $[M + H]^+$) modes. **b** *Mtb*-FbiB is a bifunctional enzyme catalyzing the reduction of dehydro-$F_{420}$-0 and its poly-glutamylation to form $F_{420}$-n. **c** Docking of $FMNH_2$ and dehydro-$F_{420}$-0 into the crystal structure of FbiB C-terminal domain. The methylene group of the enolpyruvyl moiety sits in a pocket made up of M372 and P289, while the carboxylate hydrogen bonds with R337. The methylene double bond sits planar above the isoalloxazine ring of $FMNH_2$ at an appropriate distance (3.6 Å, shown by dashed line) and oriented for a hydride transfer to the *Si* face of the methylene bond, accounting for the observed (*S*)-lactyl moiety of $F_{420}$

of archaeal genomes revealed that *cofH* and *cofG* are closely associated in genomic context[39]. We therefore investigated the genomic context of archaeal *cofE* genes to investigate whether genes with homology to the C-terminal NTR domain of FbiB were located nearby. From over 1000 archaeal genomes, we only detected 16 open reading frames (ORFs) in the neighboring context of *cofE* that could encode proteins with an NTR fold, although none of these shared substantial (>34% sequence identity) homology and all lacked the key $F_{420}$ binding residues observed in FbiB. There was one interesting exception: the unusual archaea *Lokiarchaeum* sp.[40] are unique among all sequenced archaea in that they alone encode an FbiB-like γ-glutamyl ligase:NTR fusion protein.

**Expression of *fbiABCD* is sufficient to produce $F_{420}$ in *Escherichia coli*.** Cofactor $F_{420}$ is only produced by certain bacterial species; the majority of bacteria, including *E. coli*, lack the genes required for $F_{420}$ biosynthesis. Our in vitro assay results suggest that 2-PL and, by extension, the hypothesized L-lactate kinase CofB are not required for heterologous production of $F_{420}$ in a non-native organism. To test our hypothesis, we generated a plasmid expressing the *M. smegmatis* $F_{420}$ biosynthesis genes *fbiB/C/D* along with the *fbiA* homolog *cofD* from *M. mazei*[19], which was substituted as *Ms*-FbiA was found to express poorly in *E. coli*.

We generated two versions of a plasmid-encoded recombinant $F_{420}$ biosynthetic operon, both of which encode codon-optimized genes for expression in *E. coli*: one encodes the native enzymes

($pF_{420}$), and the second encodes C-terminal FLAG-tagged versions of the enzymes to allow their detection in a western blot ($pF_{420}$-FLAG) (Fig. 5a). Plasmids were designed to allow induction of $F_{420}$ biosynthesis in the presence of anhydrotetracycline. A Western blot using anti-FLAG antibodies was used to detect whether the proteins were expressed in soluble form in *E. coli*, and confirmed that all were expressed to varying degrees (Supplementary Fig. 3). We then tested, using HPLC with fluorescence detection, whether $F_{420}$ was heterologously produced in the cell lysate of our recombinant *E. coli* strain expressing this operon. As shown in Fig. 5b, *E. coli* expressing both the FLAG-tagged and untagged plasmids produced traces consistent with mature, poly-glutamylated $F_{420}$, although they differed slightly to the *M. smegmatis*-produced $F_{420}$ standard in terms of the distribution of poly-γ-glutamate tail lengths. We also observed some accumulation of Fo (Supplementary Fig. 4), an intermediate formed by FbiC, which is also observed when $F_{420}$ is produced in *M. smegmatis*[41].

To confirm that this was indeed mature $F_{420}$, and not dehydro-$F_{420}$ species, we purified the compound from *E. coli* lysate and performed high-resolution tandem MS (MS/MS) analysis. Mass fragmentation did indeed show that the compound was reduced, and not dehydro-$F_{420}$ (Fig. 5d). No dehydro-$F_{420}$ species were detected. The yield of purified $F_{420}$ was ~27 nmol $L^{−1}$ of culture, which is comparable to physiological levels of several $F_{420}$-producing species[42]. Ultraviolet–visible (UV–Vis) and fluorescence spectra of the purified $F_{420}$ matched literature values (Supplementary Fig. 5)[4,5]. Finally, we confirmed that the purified cofactor was functional by measuring enzyme kinetic parameters with $F_{420}$-dependent glucose-6-phosphate

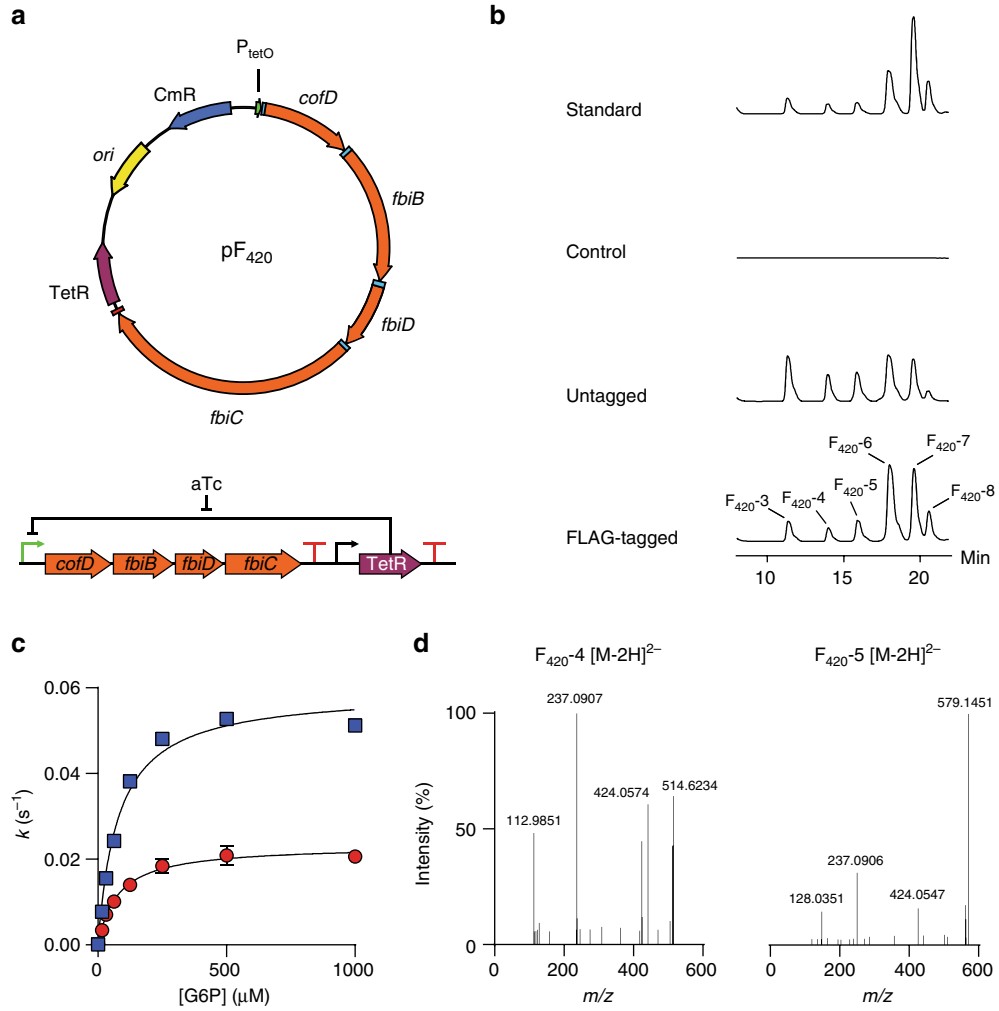

**Fig. 5** Heterologous expression of $F_{420}$ biosynthesis pathway in *E. coli*. **a** Schematic representation of the vector generated for expression of the $F_{420}$ biosynthesis pathway. **b** High-performance liquid chromatography-fluorescence detector (HPLC-FLD) traces of *E. coli* lysates containing $F_{420}$ biosynthesis constructs as well as a purified standard from *M. smegmatis*. **c** Kinetic studies indicate an identical Michaelis constant for glucose-6-phosphate dehydrogenase (FGD) as measured with $F_{420}$ purified from *M. smegmatis* (blue squares) and *E. coli* (red circles). Error bars represent standard deviations. **d** Fragmentation of $F_{420}$-4 and $F_{420}$-5 extracted from *E. coli* shows a mature $F_{420}$ production

dehydrogenase (FGD) from *M. smegmatis* (Fig. 5c). The apparent Michaelis constant was within error of that observed with FGD and $F_{420}$ produced from *M. smegmatis*, while the $k_{cat}$ was approximately half that of the *M. smegmatis* $F_{420}$ (Fig. 5c), which could result from slight differences in the distribution of tail lengths, as previously reported (Supplementary Table 2)[43]. These results confirm that the heterologous production of $F_{420}$ was achieved with a biosynthetic pathway containing only CofD (a homolog of FbiA)/FbiB/FbiC/FbiD.

Having demonstrated that our recombinant *E. coli* strain could produce mature $F_{420}$ that was functional ex vivo, we tested whether co-expression of $F_{420}$-dependent enzymes could facilitate $F_{420}$-dependent transformations in vivo. We have previously shown that $F_{420}H_2$ can decolorize malachite green nonenzymatically, and that several $F_{420}$-dependent reductases accelerate this reaction[44]. As summarized in Supplementary Table 3, *E. coli* were unable to decolorize malachite green without induction of $F_{420}$ biosynthesis. When the cells additionally expressed FGD and the $F_{420}$-dependent oxidoreductase MSMEG_2027, malachite green decolorization was significantly increased ($p < 0.001$, Student's paired $t$ test, two-sided). These results demonstrate that incorporation of the $F_{420}$ biosynthetic pathway into *E. coli* allows $F_{420}$-dependent enzymes to function in vivo upon heterologous expression.

## Discussion

It has become widely accepted within the field that one of the essential initial steps in $F_{420}$ biosynthesis involves a hypothetical L-lactate kinase that produces 2-PL, which is subsequently incorporated into $F_{420}$ through the activities of CofC and CofD. However, neither bioinformatics nor genetic knockout studies have identified plausible candidate genes for a L-lactate kinase[2,18,23]. Furthermore, 2-PL has been shown to inhibit several enzymes involved in central metabolism[28–30]. In terms of pathway flux, this makes 2-PL an unusual starting point for biosynthesis of an abundant metabolite such as $F_{420}$, which can exceed 1 µM in some species[42]. The results presented in this paper unequivocally demonstrate that PEP, rather than 2-PL, is the authentic starting metabolite in bacteria. These results reconcile the previously problematic assumptions that are required to include 2-PL within the biosynthetic pathway and establish a revised pathway (Fig. 6) that is directly linked to central carbon metabolism (via PEP) through the glycolysis and gluconeogenesis pathways.

Our observation that CofC accepts PEP (and not 2-PL) in vitro, appear to contradict previous studies in which *Mj*-CofC was reported to use 2-PL as substrate[19,35]. However, this discrepancy

is most likely due to the supplementation of the coupled CofC/CofD reaction in these studies with pyruvate kinase and 2 mM PEP, a strategy that was used to overcome apparent product

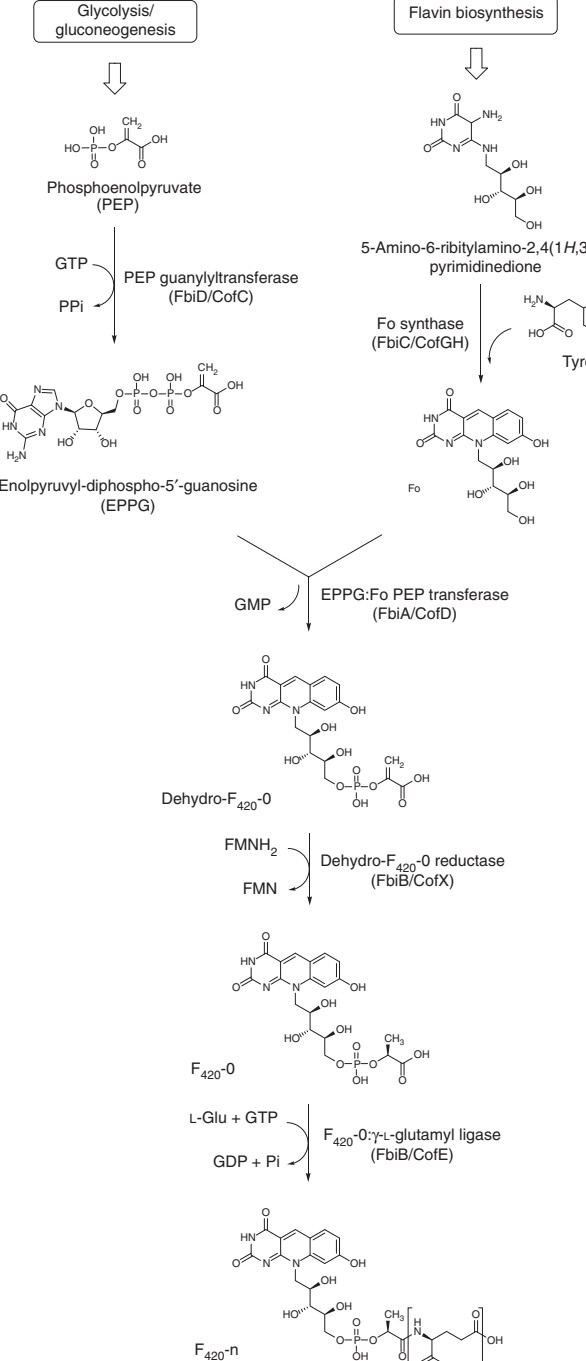

**Fig. 6** The revised bacterial $F_{420}$ biosynthesis pathway. The revised pathway is a modified scheme showing that phosphoenolpyruvate (PEP) acts as the substrate for the FbiD/CofC enzymes to produce enolpyruvyl-diphospho-5′-guanosine (EPPG) or enolpyruvyl-diphospho-5′-adenosine (EPPA) (in the case of CofC). The immediate reaction product formed from Fo and EPPG/EPPA is dehydro-$F_{420}$-0, which is reduced to $F_{420}$-0 through the newly described reductase activity of the C-terminal domain of FbiB in mycobacteria. A separate enzyme in archaea and some bacteria is expected to catalyze this reduction step (CofX). FbiB/CofE subsequently adds a poly-γ-glutamate tail to form $F_{420}$. "Fbi" refers to bacterial proteins, whereas "Cof" represents archaeal ones

inhibition by GMP[19,35]. Regardless, we cannot explain how a pathway starting from PEP can produce mature (i.e., not dehydro-$F_{420}$) $F_{420}$ in archaea given that their equivalent of FbiB (CofE) lacks the C-terminal domain dehydro-$F_{420}$-0 reductase domain seen in FbiB. One possibility is that an unknown dehydro-$F_{420}$-0 reductase exists elsewhere in the genome (remote from CofE). Further studies are required to resolve this step in archaeal $F_{420}$ biosynthesis.

The discovery of dehydro-$F_{420}$-0 as the product of FbiD:FbiA activity in mycobacteria indicated that another enzyme must be required to reduce dehydro-$F_{420}$-0 and produce $F_{420}$-0. This allowed us to define the function of the orphan C-terminal domain of mycobacterial FbiB[20]. The N-terminal domain is homologous with the archaeal γ-glutamyl ligase CofE[35], but is only capable of catalyzing glutamylation of $F_{420}$-1[20]. The C-terminal domain, which binds both $F_{420}$ and FMN, is essential for extending the poly-γ-glutamate tail of $F_{420}$-0[20], but we could not explain the possible function of the FMN cofactor. Here, we show that FbiB can catalyze both poly-glutamylation and reduction of dehydro-$F_{420}$-0, via the activities of the N- and C-terminal domains, respectively.

The increasing recognition of the importance of $F_{420}$ in a variety of biotechnological, medical, and ecological contexts underlines the need for making the compound more widely accessible to researchers; however, our inability to produce $F_{420}$ recombinantly in common laboratory organisms has been a major barrier to wider study. Here, we confirm the results of our in vitro experiments by showing that recombinant expression of the four characterized $F_{420}$ biosynthesis genes allows production of $F_{420}$ in *E. coli*. These findings should now facilitate the use of $F_{420}$ in a variety of processes with recombinant organisms, such as biocatalysis using a bio-orthogonal cofactor, directed evolution of $F_{420}$-dependent enzymes, recombinant production of antibiotics for which $F_{420}$ is a required cofactor, and metabolic engineering.

## Methods

**Bacterial strains and growth conditions**. Protein expression utilized either *M. smegmatis* mc²4517[45], *E. coli* BL21(DE3), or LOBSTR-BL21(DE3)[46] cells. For growth of *M. smegmatis*, media were supplemented with 0.05% (v/v) Tween-80. *Mycobacterium smegmatis* cells were grown in ZYM-5052[47] or modified auto-induction media Terrific Broth (TB) 2.0 (2.0% tryptone, 0.5% yeast extract, 0.5% NaCl, 22 mM $KH_2PO_4$, 42 mM $Na_2HPO_4$, 0.6% glycerol, 0.05% glucose, 0.2% lactose)[48]. For selenomethionine labeling, cells were grown in PASM-5052 media[47]. *Escherichia coli* expressions were conducted in either the above-modified auto-induction media or TB medium modified for auto-induction of protein expression (1.2% tryptone, 2.4% yeast extract, 72 mM $K_2HPO_4$, 17 mM $KH_2PO_4$, 2 mM $MgSO_4$, 0.8% glycerol, 0.015% glucose, 0.5% lactose, 0.375% aspartic acid), grown for 4 h at 37 °C followed by overnight incubation at 18 °C.

**Protein expression and purification**. *Mtb*-FbiD: The ORF encoding FbiD (Rv2983) was obtained by PCR from *M. tuberculosis* H37Rv genomic DNA (Supplementary Table 1). The pYUBDuet-*fbiABD* co-expression construct was then prepared by cloning *fbiD* into pYUBDuet[41] using *Bam*HI and *Hin*dIII restriction sites, followed by cloning the *fbiAB* operon using *Nde*I and *Pac*I restriction sites. This construct expresses FbiD with an N-terminal His₆-tag, whereas the FbiA and FbiB proteins are expressed without any tags.

The pYUBDuet-*fbiABD* vector was transformed into *M. smegmatis* mc²4517 strain[45] for expression. The cells were grown in a fermenter (BioFlo®415, New Brunswick Scientific) for 4 days before harvesting. The cells were lysed in 20 mM HEPES, pH 7.5, 150 mM NaCl, 20 mM imidazole, and 1 mM β-mercaptoethanol using a cell disruptor (Microfluidizer M-110P) in the presence of Complete protease inhibitor mixture mini EDTA-free tablets (Roche Applied Science). The lysate was centrifuged at 20,000 × g to separate the insoluble material. Recombinant FbiD was separated from other proteins by immobilized metal affinity chromatography on a HisTrap FF 5-mL Ni-NTA column (GE Healthcare), eluted with imidazole, and further purified by size-exclusion chromatography (SEC) on a Superdex 75 10/300 column (GE Healthcare) pre-equilibrated in 20 mM HEPES, pH 7.5, 150 mM NaCl, and 1 mM β-mercaptoethanol.

*Mtb*-FbiA: The ORF encoding *M. tuberculosis* FbiA was commercially synthesized and cloned into pRSET-A (Invitrogen). The pYUB28b-*fbiA* construct used for expression in *M. smegmatis* mc²4517 was prepared by

subcloning *fbiA* into pYUB28b[41] using an *Nde*I site introduced by overhang PCR utilizing the *Hin*dIII restriction site present on both vectors amplified with the T7 reverse primer (Supplementary Table 1). The resulting pYUB28b-*fbiA* construct expresses FbiA with an N-terminal His6-tag. The protein was expressed in *M. smegmatis* mc2 4517 in ZYM-5052 media auto-induction media[47,49] in a fermenter (BioFlo®415, New Brunswick Scientific) for 4 days. The protein was then purified using Ni-NTA and SEC steps, as described above, in 20 mM HEPES, pH 7.5, 200 mM NaCl, 5% glycerol, and 1 mM β-mercaptoethanol.

*Mj*-CofC: The ORF encoding *Methanocaldococcus jannaschii* CofC (MJ0887)[25] was synthesized (GenScript) and cloned into pYUB28b vector[41] using *Nde*I and HindIII restriction sites. The protein was expressed in *E. coli* in TB auto-induction media and purified using Ni-NTA and SEC steps, as described above, in 20 mM HEPES, pH 7.5, 200 mM NaCl, 5% glycerol, and 1 mM β-mercaptoethanol.

*Ec*-Fre: The *E. coli* flavin reductase[37] was cloned into pProEX-HTb using *Kas*I and HindIII restriction sites (Supplementary Table 1). Protein expression and purification was performed similar to *Mj*-CofC.

*Ms*-FbiD: The ORF encoding *M. smegmatis* FbiD (MSMEG_2392) was synthesized (Integrated DNA Technologies) and cloned into pETMCSIII by Gibson Assembly as outlined previously[48]. The protein was expressed overnight ay 30 °C in auto-induction media and purified similarly to *Mj*-CofC.

**Construction of synthetic F$_{420}$ biosynthesis operon.** Ribosome binding sites were individually optimized for each of the codon-optimized F$_{420}$ biosynthesis genes using the Ribosome Binding Site Calculator server[39,50]. Multiple operon designs were analyzed using the server's operon calculator and modified to remove unwanted translation products and RNA instability elements while maintaining predicted translation initiation rates for all coding sequences within an order of magnitude. The final design placed the operon under the control of the tetracycline-inducible promoter BBa_R0040 and the artificial terminator BBa_B1006. For subsequent assembly the operon was flanked by BioBrick prefix and suffix sequences. The operon was synthesized by GenScript and cloned into pSB1C3 containing the constitutive tetracycline repressor cassette BBa_K145201 using the standard BioBrick assembly protocol with *Eco*RI and *Xba*I/*Spe*I restriction enzymes[51]. This construct, hereafter referred to as pF$_{420}$, enables production of F$_{420}$ to be induced by the addition of anhydrotetracycline. For Western blot analysis as second version of the operon with single C-terminal FLAG tags on all four genes was likewise synthesized.

To test for in vivo F$_{420}$-depended reductase/oxidase activity, the ORF of *M. smegmatis* FGD (MSMEG_0777) was codon optimized and commercially synthesized by GenScript and cloned into multiple cloning site 1 of pCOLADuet-1 (Novagen) using *Nco*I and *Hin*dIII sites. Subsequently MSMEG_2027 was subcloned from pETMCSIII-MSMEG_2027[48] into multiple cloning site 2 using *Nde*I and *Kpn*I to give pFGD_2027.

**$Mtb$-FbiD crystallization and structure determination.** Apo-FbiD (20 mg mL$^{-1}$ in 20 mM HEPES, pH 7.5, 150 mM NaCl, 1 mM β-mercaptoethanol) was crystallized using the sitting drop vapor diffusion method in 30% polyethylene glycol 1500, 3% MPD (2-methyl-2,4-pentanediol), 0.2 M MgSO$_4$, and 0.1 M sodium acetate, pH 5.0. For experimental phasing, selenomethionine-substituted FbiD crystals were grown using protein produced in *M. smegmatis* host cells[49]. Se-SAD anomalous diffraction data sets were collected at the Australian Synchrotron. Data collection statistics are summarized in Table 1. Crystals of *Mtb*-FbiD in complex with PEP were obtained by soaking pre-formed apo crystals in precipitant solutions containing 10 mM PEP for 30 min.

All data sets were indexed and processed using XDS[52], and scaled with AIMLESS[53] from the CCP4 program suite[54]. The structure was solved using the SAD protocol of Auto-Rickshaw[55], the EMBL-Hamburg automated crystal structure determination platform. Based on an initial analysis of the data, the maximum resolution for substructure determination and initial phase calculation was set to 2.83 Å. All three of the expected heavy atoms were found using the program SHELXD[56]. The initial phases were improved using density modification and phase extension to 2.33 Å resolution using the program RESOLVE[57]. Cycles of automatic model building by ARP/wARP[58] and phenix. autobuild[59] resulted in a protein model that was completed manually using COOT[60]. Water molecules were identified by their spherical electron density and appropriate hydrogen bond geometry with the surrounding structure. Following each round of manual model building, the model was refined using REFMAC5[61], against the data to 1.99 Å resolution. The PDB_redo program[62] was used in the final stages of refinement. Full refinement statistics are shown in Table 1.

The structure of *Mtb*-FbiD in complex with PEP was solved by molecular replacement using PHASER[63] with the apo-FbiD structure as a search model. The structure was refined by cycles of manual building using COOT[60] and refinement using REFMAC5[61], against the data to 2.18 Å resolution. Full refinement statistics are shown in Table 1. The Ramachandran statistics analyzed using MolProbity[64] are 99.18% favored, 0.82% allowed, and 0.0% outliers for apo-FbiD and 99.02% favored, 0.98% allowed, and 0.0% outliers for PEP-FbiD.

Multiple sequence alignments were performed using Expresso from the T-Coffee suite[65] with the structure of *Mtb*-FbiD. Figures were prepared using the ESPript 3.0 web server[66].

**HPLC assays.** FbiD/CofC-FbiA-coupled activity was monitored in a reaction mixture containing 100 mM HEPES, pH 7.5, 2 mM GTP, 0.1 mM Fo, 5 mM MgCl$_2$, 1 mM 2-PL or PEP, 1 μM FbiD, and 5 μM FbiA. The reactions were incubated at 37 °C and stopped using 20 mM EDTA at various time points. Separation of F$_{420}$ species was performed on an Agilent HP 1100 HPLC system equipped with photodiode array and fluorescence detectors (Agilent Technologies). Samples were kept at 4 °C, and the injection volume was 20 μL. Samples were separated on a Phenomenex Luna C18 column (150 × 3 mm$^2$, 5 μm) with a 0.2 μm in-line filter that was maintained at 30 °C. The mobile phase consisted of 100% methanol (A) and 25 mM sodium acetate buffer, pH 6.0 (B), with a gradient elution at a flow rate of 0.5 mL min$^{-1}$ and a run time of 30 min. The gradient profile was performed as follows: 0–25 min 95–80% B, 25–26 min 80% B, 26–27 min 95% B, 27–30 min 95% B, and a post-run of 2 min. The wavelengths used for photodiode array were 280 and 420 nm (bandwidth 20 nm) using a reference of 550 nm (bandwidth 50 nm). The wavelengths used for the fluorescence detector were 420 nm (excitation) and 480 nm (emission).

**LC-MS characterization of dehydro-F$_{420}$ species.** Enzymatic reactions were set up as described above. Ten microliters aliquots were injected onto a C18 trap cartridge (LC Packings, Amsterdam, The Netherlands) for desalting prior to chromatographic separation on a 0.3 × 100 mm$^2$ 3.5 μm Zorbax 300SB C18 Stablebond column (Agilent Technologies, Santa Clara, CA, USA) using the following gradient at 6 μL min$^{-1}$: 0–3 min 10% B, 24 min 50% B, 26 min 97% B, 29 min 97% B, 30.5 min 10% B, 35 min 10% B, where A was 0.1% formic acid in water and B was 0.1% formic acid in acetonitrile. The column eluate was ionized in the electrospray source of a QSTAR-XL Quadrupole Time-of-Flight mass spectrometer (Applied Biosystems, Foster City, CA, USA). For IDA (Information Dependent Analysis) analyses, a TOF-MS scan from 330–1000 *m/z* was performed, followed by three rounds of MS/MS on the most intense singly or doubly charged precursors in each cycle. For targeted work, defined Product Ion Scans were created to isolate and fragment specific ions of interest with various collision energies (10–60 kV). Both positive and negative modes of ionization were used as appropriate.

**MS/MS confirmation of reduction of dehydro-F$_{420}$ species.** To confirm the reduction of the PEP moiety in vitro assays were prepared in 50 mM HEPES pH 7.5, 100 mM KCl, 5 mM MgCl$_2$, 2 mM GTP, 0.1 mM Fo, 1 mM PEP, 5 μM *Mtb*-FbiA, 6.5 μM *Msg*-FbiD, 10 mM DTT, 20 μM FMN, 0.2 mM NADH, 0.1 μM *Ec*-Fre, 2 μM *Mtb*-FbiB and 1 mM L-glutamate. To minimize futile oxi-dation of FMNH$_2$ by oxygen the reaction mixture was repeatedly evacuated and purged with nitrogen and maintained under a nitrogen atmosphere. Samples were incubated at ambient temperature for up to 36 h and stopped by addition of 20 mM EDTA. Samples were desalted using Bond Elut C18 tips (Agilent Technologies) and eluted in 0.1% formic acid in acetonitrile. Samples were injected onto a Q Exactive Plus (Thermo Fisher Scientific) at 100 μLmin$^{-1}$ with isocratic 50% B, where A was 10 mM ammonium acetate, pH 6.0, and B was 0.02% ammonia in methanol. Scans from 150–2250 *m/z* were performed and data-dependent MS/MS on targeted metabolites was done in both positive and negative modes.

**Docking dehydro-F$_{420}$-O into FbiB C-terminal domain.** Structures of FbiB C-terminal domain bound with F$_{420}$-0 (4XOQ) and FMN (4XOO) were aligned in PyMOL and the ligands removed from the structures. Chemical dictionaries of dehydro-F$_{420}$-0 and FMNH$_2$ were prepared with AceDRG[67] and built into the same molecule with COOT[60] using the two density maps as appropriate. Dehydro-F$_{420}$-0 was reparametrized using the OPLS3e forcefield[38] in Maestro (Schrödinger Release 2018-4: Maestro, Schrödinger, LLC, New York, NY, 2018). The energy was minimized using the energy minimization function of Desmond (Schrödinger Release 2018-4: Desmond Molecular Dynamics System, D. E. Shaw Research, New York, NY, 2018; Maestro-Desmond Inter-operability Tools, Schrödinger, New York, NY, 2018).

**Production of F$_{420}$ in *Escherichia coli*.** The pF$_{420}$ vector was transformed into *E. coli* BL21(DE3) for expression. Cells were cultivated in M9 minimal media supplemented with chloramphenicol (25 μg mL$^{-1}$) and tyrosine (18 μg mL$^{-1}$) in 1 L shake flasks with 500 mL working volume at 28 °C. At an optical density (OD) of 0.60, 200 ng mL$^{-1}$ of tetracycline was added to induce the expression of F$_{420}$ biosynthesis pathway. After induction, cells were cultivated for at least 16 h. To test the expression of the F$_{420}$ biosynthetic genes, cell pellets were lysed by resuspending to OD of 0.8 in buffer containing BugBuster (Novagen). Following centrifugation at 16,000 × *g*, proteins were resolved on a gel (Bolt™ 4–12% Bis-Tris Plus, Invitrogen) for 1 h and visualized using Coomassie Brilliant Blue. For immunoblotting, proteins were transferred to a membrane (iBlot® 2 NC Regular Stacks, Invitrogen) using iBlot Invitrogen (25 V, 6 min). After staining and de-staining, the membrane was blocked with 3% skim milk

solution and blotted with anti-FLAG antibodies conjugated with HRP (DYKDDDDK Tag Monoclonal Antibody, Thermo Fisher Scientific, MA1-91878-HRP, ×1000 dilution).

For detection of $F_{420}$ in *E. coli* lysate by HPLC-FLD, cells were grown overnight in media containing 2.0% tryptone, 0.5% yeast extract, 0.5% NaCl, 22 mM $KH_2PO_4$, 42 mM $Na_2HPO_4$, 100 ng mL$^{-1}$ anhydrotetracycline, and 34 μg mL$^{-1}$ chloramphenicol at 30 °C. Cells from 500 μL of culture were pelleted by centrifugation at 16,000 × *g*. Cells were re-suspended in 500μL of 50 mM $Na_2HPO_4$, pH 7.0, and lysed by boiling at 95 °C for 10 min. Cell debris was pelleted by centrifugation at 16,000 × *g* and filtered through a 0.22 μm PVDF filter. Analysis was conducted as described previously[2]. pSB1C3 containing only BBa_K145201 was used as a control.

**Purification and analysis of *Escherichia coli*-produced $F_{420}$.** For $F_{420}$ extraction, 1 L of cell culture was centrifuged at 5000 × *g* for 15 min. The cell pellet was re-suspended in 30 mL of 75% ethanol and boiled at 90 °C in a water bath for 6 min for cell lysis. The cell extract was again centrifuged at 5000 × *g* for 15 min to remove cell debris and the supernatant was lyophilized. The lyophilized cell extract was re-dissolved in 10 mL milli Q water centrifuged at 5000 × *g* for 15 min and the supernatant passed through 0.45 μm syringe filter (Millex-HV). The filtered cell extract was purified for $F_{420}$ using a 5 mL HiTrap QFF column (GE Healthcare) as previously described[41]. The purified $F_{420}$ solution was further desalted by passing it through C18 extract column (6 mL, HyperSep C18 Cartridges, Thermo Fisher Scientific). The C18 extract column was first equilibrated by passing through 10 mL of 100% methanol and 10 mL of milli Q. Afterwards, the $F_{420}$ solution was passed through the C18 extract column, and $F_{420}$ was eluted in 2 mL fraction of 20% methanol. The purified $F_{420}$ solution was further concentrated in GeneVac RapidVap and re-dissolved in 500 μL of milli Q for further analysis and assays. To detect Fo accumulation in the media, it was filtered through a 0.45 μm syringe filter and purified directly on the C18 extract column as above.

UV and fluorescence spectra were collected on a Varian Cary 60 and a Varian Cary Eclipse, respectively, in a 10 mm QS quartz cell (Hellma Analytics). Samples were buffered to pH 7.5 with 50 mM HEPES and scanned from 250 to 600 nm. For fluorescence, the excitation wavelength was 420 nm and the emission scanned from 435 to 600 nm.

Activity assays with *M. smegmatis* and *E. coli*-derived $F_{420}$ were conducted with *M. smegmatis* FGD expressed and purified as described previously[15]. Assays were performed in 50 mM Tris-HCl, pH 7.5, 300 mM NaCl, 50 nM FGD, 5 μM $F_{420}$, and 0–1000 μM glucose-6-phosphate using a SpectraMAX e2 plate reader. Activity was measured by following loss of $F_{420}$ fluorescence at 470 nm. Apparent $k_{cat}$ and $K_m$ values were calculated using the GraphPad Prism 7.04 (GraphPad Software, La Jolla, CA, USA).

For malachite green reduction assays, cells were grown in 50 mL Falcon tubes containing 30 mL M9 minimal media, and supplemented with appropriate antibiotics at 28 °C. Cells were induced at $OD_{600}$ of 0.6 by the addition of 200 ng μL$^{-1}$ tetracycline and 0.2 mM isopropyl β-D-1-thiogalactopyranoside. Cells were grown for an additional 6 h before adding malachite green to an initial $OD_{615}$ of 0.26. Cultures were covered in foil and grown in the dark to prevent spontaneous decolorization. After overnight incubation, OD at 615 nm was measured with assays conducted in triplicate. Statistical analysis was performed using GraphPad Prism 7.04 (GraphPad Software, La Jolla, CA, USA).

**2-PL synthesis.** In the absence of a commercial source, 2-PL was chemically synthesized by a slight modification of the method of Ballou and Fischer[68]. Briefly, benzyl lactate was condensed with chlorodiphenyl phosphate in pyridine, with cooling, to give benzyldiphenylphosphoryl lactate. Hydrogenolysis of this material in 70% aqueous tetrahydrofuran over 10% Pd-C gave phospholactic acid as a colorless, viscous oil, which was characterized by proton, carbon, and phosphorus nuclear magnetic resonance (NMR) spectroscopy, and by MS. $^1H$ NMR (DMSO-$d_6$) δ 11.68 (br, 3H), 4.53 (m, 1H), 1.36 (d, *J* = 6.8 Hz, 3H). $^{13}C$ NMR (DMSO-$d_6$) δ 172.50 (d, $J_{P-C}$ = 0.05 Hz), 69.56 (d, $J_{P-C}$ = 0.04 Hz), 19.27 (d, $J_{P-C}$ = 0.04 Hz). $^{31}P$ NMR (DMSO-$d_6$) δ −1.64. APCI-MS found: $[M + H]^+$ = 171.1, $[M − H]^-$ = 169.1.

**Fo purification.** Fo was purified from *M. smegmatis* culture medium over-expressing *Mtb*-FbiC as described previously[41].

**Genomic context analysis.** A non-redundant CofE dataset of 4813 sequences was collected using the Pfam identifier PF01996 and the InterPro classifications IPR008225, IPR002847, and IPR023659. Archaeal CofE sequences were extracted from this dataset, resulting in a set of 1060 sequences that included representatives across 12 phyla: Crenarchaeota, Euryarchaeota, Thaumarchaeota, *Candidatus* Bathyarchaeota, *Candidatus* Diapherotrites, *Candidatus* Heimdallarchaeota, *Candidatus* Korarchaeota, *Candidatus* Lokiarchaeota, *Candidatus* Marsarchaeota, *Candidatus* Micrarchaeota, *Candidatus* Odinarchaeota, and *Candidatus* Thorarchaeota. The genomic context (10 upstream and 10 downstream genes) of each archaeal *cofE* was analyzed for

the presence of a neighboring PF00881 domain with homology to the C-terminal domain of FbiB using the Enzyme Function Initiative—Genome Neighborhood Tool[69].

**Reporting Summary.** Further information on experimental design is available in the Nature Research Reporting Summary linked to this article.

## Data availability

The coordinates and structure factors for the PEP-bound and apo structures of Rv2983 have been deposited in the Protein Data Bank under the accession codes "6BWH" and "6BWG," respectively. All other data are available from the corresponding authors upon request.

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

## Acknowledgements

We thank Assoc. Prof. Chris Squire, Dr. Carol Hartley, Dr. Andrew Warden, and Dr. Matthew Taylor for helpful discussions, Dr. Adam Carroll and Dr. Thy Truong for technical assistance. This research is supported by a Sir Charles Hercus Fellowship (G.B.), an NHMRC New Investigator Grant (C.G.; 1142699), an ARC DECRA Fellowship (C.G.; DE170100310), the Health Research Council of New Zealand (E.N.B.), an AGRTP Scholarship (J.A.), the Australian Research Council, and the National Health and Medical Research Council (C.J.J.). Crystal data collection was undertaken on the MX1 beamline at the Australian Synchrotron (Clayton, VIC, Australia). Access to the Australian Synchrotron was supported by the New Zealand Synchrotron Group Ltd.

## Author contributions

J.A. performed experiments, analyzed results, and co-wrote the manuscript; E.N.M.J., M. V.S., J.C., S.M.S., S.S., B.P., M.M., and N.T. performed experiments and analyzed results; B.N. conceived project, performed experiments, and analyzed results; C.G. and E.N.B. conceived project, designed experiments, analyzed results, and co-wrote the manuscript; C.S. designed experiments, analyzed results, and co-wrote the manuscript; G.B. and C.J.J. conceived project, designed experiments, performed experiments, analyzed results, and co-wrote the manuscript. All authors provided feedback on the manuscript.

## Additional information

**Competing interests:** The authors declare no competing interests.

