## [Peer Review File · Nature Communications]

Reviewers' comments:

Reviewer #1 (Remarks to the Author):

A revised pathway is presented for the synthesis of coenzyme F420 in the domain Bacteria that features the characterization of two enzymes not previously described. The results have led to heterologous expression of the pathway in *Escherichia coli*. Although expression yields functional coenzyme F420, it is not a breakthrough in obtaining critical amounts for research as claimed by the authors. F420 is produced in similar amounts in methylotrophic methanogens that are relatively facile to culture in large scale. The significance of the work is of less practical value and more of the characterization of two enzymes in the bacterial pathway yet leaving gaps in the archaeal pathway.

The work is performed well and the manuscript written in a concise and lucid manner.

Reviewer #2 (Remarks to the Author):

A revised biosynthetic pathway for the cofactor F420 in bacteria
Bashiri and Antoney, et. al.

The authors report an amended biosynthetic pathway for cofactor F420 in which phosphoenolpyruvate, rather than 2-phospho-L-lactate, is a key intermediate. They also demonstrate for the first time that FbiB, a F420-biosynthetic enzyme, is bifunctional, with both γ -glutamyl ligase and F420 reductase activity. Their conclusions are supported by thorough enzymatic assays and good structural analyses. These studies have collectively enabled the heterologous expression of F420 in *E. coli*. The structural work presented is of good quality. The findings represent important contributions to the field and to bacterial metabolism more broadly, as this cofactor is involved in many critical processes. I have only minor suggestions.

(1) The structural findings of the work are not discussed in the abstract or introduction. These sections should include a brief description of the structures obtained and docking studies performed, as well as the support they provide to the authors' conclusions.

(2) A surface representation of the PEP binding site would be helpful. The text describes the binding cleft as "cone-shaped" with "a groove running across the base of the cone" and references Fig. 2a; however, the shape of the binding pocket is difficult to appreciate in the cartoon representation shown in Fig 2a.

(3) Fig. 2c should be referenced in the text on p. 7, paragraph 1, line 10 and not on p. 7, paragraph 2, line 7.

(4) In the figure legend for Fig. 2, remove the sentence beginning "The protruding hairpin in the Mj-CofC structure...", add it to the text, and provide a reference for this statement. Also, provide the PDB ID for Mj-CofC in the abstract.

(5) Are the PEP binding residues (particularly the DDD motif) more broadly conserved among FbiD/CofC orthologs from F420-producing bacteria? It would be helpful to see a sequence alignment, which could be easily added as a panel to Fig. S2.

(6) Can the authors comment on the structural basis for FbiD selectivity of PEP over 2-PL based on their structure? Perhaps analysis of GTP binding in *E. coli* MobA could provide additional insights. Further discussion in the text would be helpful, and perhaps additional structure figures.

(7) Is there any evidence that FbiD/CofC forms a complex with FbiA/CofD?

(8) Switch the labels for Fig 3b and Fig 3c so they are consistent with how they are introduced in the text.

(9) As described in point (2), a surface representation of Fig 3b would help the reader visualize the hydrophobic pocket formed by P289 and M375 and other structural features described in the text.

(10) On p. 9, paragraph 2, line 3, please reference the structures of FbiB bound to FMN and F420, respectively ("Crystal structures with both F420 and FMN bound have been separately solved...").

Reviewer #3 (Remarks to the Author):

The paper describes research undertaken that demonstrates how cofactor F420 is made in bacteria and methanogens. The authors show, convincingly, that PEP is a substrate in the reaction, which is combined with GTP to generate a likely but transient intermediate termed EPPG (not EGGP as described on page 6) that is converted into dehydro F420. Furthermore, the authors also demonstrate that dehydro F420 is converted into F420-O in an FMNH₂-dependent manner. These represent new discoveries within the area and thereby clarify how cells are able to make cofactor F420 through the action of 4 proteins. This was further confirmed by producing these enzymes recombinantly in *E. coli*, which resulted in the production of the cofactor within the host, a compound it cannot normally make. This work makes a significant contribution to the field.

There are a few points the authors may wish to consider.

1. The title could be mis-interpreted. I would go for a simpler statement - A revised biosynthesis for cofactor F420 in prokaryotes.
2. Abstract - final sentence - expression of F420 from a
3. Introduction - I personally would like to see a Figure in this section outlining the branches of the pathway.
4. One page 6 - to what extent have the authors tried to isolate and confirm the structure of EPPG? Can the structure of dehydro F420 be confirmed further by NMR? Although the latter has been confirmed by MS, more analytical data could make this even more convincing.
5. Page 9. What is the OP3a forcefield - I think some kind of description and reference is required.
6. The level of recombinant biosynthesis seems quite low. Where any intermediates observed? Can the recombinant strain producing F420 be made to co-express a F420-dependent enzyme - and does this increase yield?

Reviewer #2 (Remarks to the Author):

A revised biosynthetic pathway for the cofactor F420 in bacteria

Bashiri and Antoney, et. al.

The authors report an amended biosynthetic pathway for cofactor F420 in which phosphoenolpyruvate, rather than 2-phospho-L-lactate, is a key intermediate. They also demonstrate for the first time that FbiB, a F420-biosynthetic enzyme, is bifunctional, with both γ -glutamyl ligase and F420 reductase activity. Their conclusions are supported by thorough enzymatic assays and good structural analyses. These studies have collectively enabled the heterologous expression of F420 in *E. coli*. The structural work presented is of good quality. The findings represent important contributions to the field and to bacterial metabolism more broadly, as this cofactor is involved in many critical processes. I have only minor suggestions.

(1) The structural findings of the work are not discussed in the abstract or introduction. These sections should include a brief description of the structures obtained and docking studies performed, as well as the support they provide to the authors' conclusions.

We have modified the abstract (lines 9,10) to include mention of the structure. There is no space in the abstract to mention the docking (we are already 9 words over the limit). However, we have added discussion of both the crystal structure and the docking in the last paragraph of the introduction (lines 59-67).

(2) A surface representation of the PEP binding site would be helpful. The text describes the binding cleft as "cone-shaped" with "a groove running across the base of the cone" and references Fig. 2a; however, the shape of the binding pocket is difficult to appreciate in the cartoon representation shown in Fig 2a.

We have now included a molecular surface representation (Figure 3a) to illustrate the groove. The figure is referenced on line 118.

(3) Fig. 2c should be referenced in the text on p. 7, paragraph 1, line 10 and not on p. 7, paragraph 2, line 7.

This is now fixed. Line 116.

(4) In the figure legend for Fig. 2, remove the sentence beginning "The protruding hairpin in the Mj-CofC structure..." add it to the text, and provide a reference for this statement. Also, provide the PDB ID for Mj-CofC in the abstract.

Line 114. We have moved the sentence to the text. There is no publication for the structure, so we have cited the PDB entry (DOI: 10.2210/pdb2I5E/pdb).

(5) Are the PEP binding residues (particularly the DDD motif) more broadly conserved among FbiD/CofC orthologs from F420-producing bacteria? It would be helpful to see a sequence alignment, which could be easily added as a panel to Fig. S2.

Lines 121-122. We have added the alignment to Figure S2 as requested. The motif is fully conserved.

(6) Can the authors comment on the structural basis for FbiD selectivity of PEP over 2-PL based on their structure? Perhaps analysis of GTP binding in *E. coli* MobA could provide additional insights. Further discussion in the text would be helpful, and perhaps additional structure figures.

We have added an explanation for the selectivity of PEP over 2-PL (lines 126-129). This has been studied in depth in other PEP-related enzymes (Nowak and Mildvan (1970), JBC, 245, 6057), and relates to the different stereochemistry affecting binding of the other substrate or ability to attain a productive transition state.

(7) Is there any evidence that FbiD/CofC forms a complex with FbiA/CofD?

Despite many attempts, we could not obtain a stable complex, so we assume it is transient and only forms during the reaction. We make it clear that we speculate that it forms a complex based on the lack of activity of FbiD in the absence of FbiA, and the fact that we could not detect binding of GTP to FbiD (in the absence of FbiA). We also solved more than a dozen structures of FbiD in the presence of various NTPs in different conditions and never observed any indication of binding. So, the evidence is indirect, which is why we make it clear it is speculation. We hope this is acceptable.

(8) Switch the labels for Fig 3b and Fig 3c so they are consistent with how they are introduced in the text.

This has been fixed. Lines 161, 175.

(9) As described in point (2), a surface representation of Fig 3b would help the reader visualize the hydrophobic pocket formed by P289 and M375 and other structural features described in the text.

Line 178. We have provided a molecular surface representation (Figure 4c), to help illustrate the hydrophobic pocket formed by P289, M375, etc.

(10) On p. 9, paragraph 2, line 3, please reference the structures of FbiB bound to FMN and F420, respectively ("Crystal structures with both F420 and FMN bound have been separately solved...").

Lines 164, 165. We have now included the PDB IDs and reference for these structures.

Reviewer #3 (Remarks to the Author):

The paper describes research undertaken that demonstrates how cofactor F420 is made in bacteria and methanogens. The authors show, convincingly, that PEP is a substrate in the reaction, which is combined with GTP to generate a likely but transient intermediate termed EPPG (not EGPP as described on page 6) that is converted into dehydro-F420. Furthermore, the authors also demonstrate that dehydro-F420 is converted into F420-O in an FMNH₂-dependent manner. These represent new discoveries within the area and thereby clarify how cells are able to make cofactor F420 through the action of 4 proteins. This was further confirmed by producing these enzymes recombinantly in *E. coli*, which resulted in the production of the cofactor within the host, a compound it cannot normally make. This work makes a significant contribution to the field.

There are a few points the authors may wish to consider.

1. The title could be mis-interpreted. I would go for a simpler statement - A revised biosynthesis for cofactor F420 in prokaryotes.

This change has been made.

2. Abstract - final sentence - expression of F420 from a

Line 14, this change has been made. We also corrected heterologous expression to heterologous production.

3. Introduction - I personally would like to see a Figure in this section outlining the branches of the pathway.

Line 31. We have now included a new Figure 1, which outlines the branches in the currently accepted biosynthesis pathway.

4. One page 6 - to what extent have the authors tried to isolate and confirm the structure of EPPG? Can the structure of dehydro F420 be confirmed further by NMR? Although the latter has been confirmed by MS, more analytical data could make this even more convincing.

Like others working on LPPG (Graupner & White (2001) *Biochemistry* 40, 10859), we found that EPPG was simply too unstable to isolate and its structure needed to be inferred. We made many attempts but were not successful.

Regarding dehydro-F₄₂₀, we also tried to confirm the results using NMR, however this was unsuccessful because of three main reasons. First, the reaction is slow and relatively inefficient; the

yield in the coupled enzyme reaction was very low. Second, the low yield that we could obtain was reduced further because of the need to purify the species from the complex reaction mixture into a buffer suitable for NMR. Third, we have observed rapid decomposition of dehydro-F₄₂₀-O into a compound with m/z 424, which is likely the cyclic phosphodiester of Fo. This rapid decomposition was the primary reason we opted for a laborious four-enzyme coupled assay. However, although NMR would make the argument more convincing, the high-resolution tandem mass spectrometry results we present in the paper are extremely sensitive (within 5 ppm) and make the assignment and conclusion unambiguous, especially when viewed in the context of the additional structural and kinetic results.

5. Page 9. What is the OPLS3a forcefield - I think some kind of description and reference is required.

Lines 169-173. We apologise, the forcefield was originally misspelled. We have corrected this to OPLS3e and added a reference to the paper. We opted to use the OPLS3e forcefield as it has a substantially greater number of rotational and torsional parameters compared to other forcefields available, and provides more accurate descriptions of the potential energy surfaces of ligands, leading for improved docking poses over other forcefields. It is the most advanced forcefield available for this type of analysis.

6. The level of recombinant biosynthesis seems quite low. Where any intermediates observed? Can the recombinant strain producing F₄₂₀ be made to co-express a F₄₂₀-dependent enzyme - and does this increase yield?

Lines 216, 217. We have observed Fo in the media of cultures expressing the biosynthetic genes. Fo also accumulates in the culture media during production of F₄₂₀ in *M. smegmatis*. We have included a SI Figure showing this, and cited the *M. smegmatis* paper on line 216. We also detect different polyglutamylated forms, as we already mention. Beyond this, we could not detect any other species. We suspect they are produced in very low amounts and quickly converted to the next product in the pathway.

Lines 232-241. The yield is significant, within the ranges observed naturally, and allows for purification and detection of the cofactor. This is the first time this has been achieved, so we expect that we will be able to improve in the future. We have not observed an increase in yield from co-expression of F₄₂₀-dependent enzymes. However, we have added new data (Table S4), showing that F₄₂₀-dependent enzymes are active in vivo in the recombinant F₄₂₀-producing strain.

REVIEWERS' COMMENTS:

Reviewer #3 (Remarks to the Author):

All the comments that I made previously have been addressed. Reads well - good paper and makes a significant contribution to the field.